# Temporal SNR optimization through RF coil combination in fMRI: The more, the better?

Redouane Jamil[1], Franck Mauconduit[1], Caroline Le Ster[1], Philipp Ehses[2], Benedikt A. Poser[3], Alexandre Vignaud[1], Nicolas Boulant[1]*

**1** CEA, CNRS, BAOBAB, NeuroSpin, Paris-Saclay University, Gif-sur-Yvette, France, **2** German Center for Neurodegenerative Diseases (DZNE), Bonn, Germany, **3** Department of Cognitive Neuroscience, Maastricht Brain Imaging Centre, Faculty of Psychology and Neuroscience, Maastricht University, Maastricht, The Netherlands

* nicolas.boulant@cea.fr

**Data Availability Statement:** doi of the data set: 10.18112/openneuro.ds003777.v1.0.1.

**Funding:** Leducq foundation large equipment ERPT program,NEUROVASC7T project https://www.fondationleducq.org/ European Union's Horizon

## Abstract

For functional MRI with a multi-channel receiver RF coil, images are often reconstructed channel by channel, resulting into multiple images per time frame. The final image to analyze usually is the result of the covariance Sum-of-Squares (covSoS) combination across these channels. Although this reconstruction is quasi-optimal in SNR, it is not necessarily the case in terms of temporal SNR (tSNR) of the time series, which is yet a more relevant metric for fMRI data quality. In this work, we investigated tSNR optimality through voxel-wise RF coil combination and its effects on BOLD sensitivity. An analytical solution for an optimal RF coil combination is described, which is somewhat tied to the extended Krueger-Glover model involving both thermal and physiological noise covariance matrices. Compared experimentally to covSOS on four volunteers at 7T, the method yielded great improvement of tSNR but, surprisingly, did not result into higher BOLD sensitivity. Solutions to improve the method such as for example the t-score for the mean recently proposed are also explored, but result into similar observations once the statistics are corrected properly. Overall, the work shows that data-driven RF coil combinations based on tSNR considerations alone should be avoided unless additional and unbiased assumptions can be made.

## Introduction

The blood oxygenation level-dependent (BOLD) functional MRI (fMRI) contrast increases with magnetic field strength [1], but still represents only a few percent of signal change. Thermal noise, head motion, scanner instabilities and a variety of physiological phenomena such as breathing and cardiac pulsations potentially make the signal changes due to neural activations hard to detect reliably.

Current state of the art in MRI acquisition uses multi-receive channel RF coils to increase SNR [2] and benefit from parallel imaging [3, 4]. In accelerated acquisitions that utilize GRAPPA, images from each receiver channel are often reconstructed individually, resulting in a multitude of images per time frame that are generally combined during reconstruction. The most standard coil combination to reconstruct a single image per time frame is the root Sum-

2020 research and innovation program (AROMA project) Grant no 885876 https://aroma-h2020.com/ NO - The funders had no role in study design, data collection and analysis, decision to publish, or preparation of the manuscript.

**Competing interests:** The authors have declared that no competing interests exist.

of-Squares (SoS), due to its convenient implementation and SNR quasi-optimality [5]. A pre-whitening can be done at this stage for further performance by inserting the thermal noise covariance matrix computed from noise pre-scans [2, 6], a method labelled here covSoS. While solidly motivated by the theory in the thermal noise regime, the method however is not necessarily optimal in terms of temporal SNR (tSNR), indicative of signal stability and thus of more relevance for fMRI data quality. It has been well known that according to the model of Krueger and Glover [7], non-thermal sources of noise such as physiological noise and scanner instabilities can lead to a plateau of tSNR despite SNR boosts, even on phantoms [1]. When benchmarking sequences with pilot tests before applying an fMRI paradigm on a cohort, as a result the temporal aspect of the signal or, ideally, the neural activations themselves should always be taken into account [8]. Computed from the ratio of the activation spike amplitude over the standard deviation of the noise time-series, the functional Contrast to Noise Ratio (fCNR) is conceptually a very accurate metric to measure the quality of a task-based fMRI acquisition. However, because the location of the activations and their strengths are in theory unknown, tSNR has remained arguably one of the most popular metrics to guide the experimenter's choices [9].

tSNR mathematically consists of assessing signal stability through time for each voxel via the ratio of the mean to the standard deviation of the time-series. But because temporal-correlations may also exist [10], this metric has also been shown to be less well correlated with t-scores than the t-score for the mean, which in essence is the same as tSNR but after GLM analysis and thus taking temporal correlations into account [11]. Moreover, although the tSNR has already been shown to not correlate well with t-scores [12, 13], it is still in general believed that the t-score versus tSNR relationship is an increasing function [11, 14].

Alternative coil combinations have already been proposed to specifically improve tSNR. Drawing the parallel with the thermal noise covSOS approach, Triantafyllou et al. [15] suggested considering the use of the time-series noise covariance matrix $\Psi_t$ to account also for the physiological noise. Likewise, Huber et al. [16, 17] proposed the STAbility weighted RF coil Combination (STARC) method consisting in a voxel wise tSNR optimized weighted sum of channels. Initially solved through a gradient descent method and thus hardly implementable into online reconstruction, we provide here an analytical solution to the problem, which turns out to exploit the total noise covariance matrix $\Psi_t$ from Triantafyllou et al. [15]. Variants of this solution are also explored; the first consisted simply of optimizing the weights based on a pre-scan and then applying them to the fast-event fMRI scans. The second consisted of directly optimizing the t-score for the mean [10], which after GLM analysis, aimed at filtering out the activations while maximizing signal stability so that activations did not influence the computation of the weights. The revised STARC method and its variants are compared to covSoS through fMRI experiments on four healthy volunteers at 7T. Temporal SNR, activation maps, and scatter plots linking tSNR gains versus t-score gains are computed and compared. Example of optimization results with bar graphs and signal time courses are also provided to illustrate further the behaviour of STARC compared to covSoS.

## Theory

Throughout this work, for a given image voxel $K$, we denote by $S_K(n)$ its $N_c \times 1$ signal vector at the volume repetition $n$ through all receive channels with $N_c$ the number of receive channels and $A$ the matrix of dimension $N \times N_c$ concatenating the signal time courses from all coils of

that voxel: $A = \begin{pmatrix} S_K(1)^H \\ \vdots \\ S_K(N)^H \end{pmatrix}$. The tSNR is expressed as $tSNR = \frac{mean(I)}{std(I)}$ where $mean(I)$ is the

temporal mean of image voxel $K$, $std(I)$ its temporal standard deviation while $I$ is the resulting voxel intensity after coil combination. The superscript $H$ denotes Hermitian conjugate.

The SoS and covSoS combinations have their image voxel intensity computed as

$$I_{SoS}(n) = \sqrt{S_K(n)^H S_K(n)}, \qquad I_{covSoS}(n) = \sqrt{S_K(n)^H \Psi_0^{-1} S_K(n)}$$

The current gold standard is the covSoS approach where the $N_c \times N_c$ channel covariance matrix $\Psi_0$ computed from noise only pre-scans (scans without RF) is inserted between signal vectors. When $\Psi_0^{-1}$ is decomposed into its Cholesky form, this operation can be seen as a pre-whitening process uncorrelating the channels and penalizing the noisiest ones.

In [15], Triantafyllou et al. extended the Krueger-Glover model by substituting to the scalar coefficient expressing the effective strength of the physiological noise with a physiological noise covariance matrix $\Psi_p$ such that the time-series covariance matrix $\Psi_t$, calculated from the covariance matrix of $A$, could be separated into $\Psi_t = \Psi_0 + \Psi_p$. The paper concluded by suggesting that $\Psi_t$ could thereby be leveraged in the coil combination to optimize tSNR, just as the covSOS approach does for SNR, as intuition would dictate:

$$I_{covSoS_t}(n) = \sqrt{S_K(n)^H \Psi_t^{-1} S_K(n)} \text{ with } \Psi_t = cov(A).$$

Originally introduced by Huber et al., STARC [16] is a voxel-wise data-driven tSNR optimization yielding a weighted sum of channels written as $I_{STARC} = AX$ with $X$ the $N_c \times 1$ coil combination vector to determine for each voxel. The STARC problem was originally written as

$$X = \underset{X}{\operatorname{argmax}} \frac{mean(I_{STARC})}{std(I_{STARC})}$$

The optimization had to be performed on each voxel independently via a gradient descent method and was thus time-consuming. However, it is possible to provide an analytical solution to this problem by recasting it as

$$X = \underset{X}{\operatorname{argmin}} \; Var(I_{STARC}) \; s.t. E(I_{STARC}) = b,$$

where $E(I_{STARC})$ and $Var(I_{STARC})$ denote respectively the expectation and the variance values of the time signal $I_{STARC}$. Their respective expressions are

$$E(I_{STARC}) = u'X, \quad Var(I_{STARC}) = \frac{1}{N} X'(A'A - uu')X = X'cov(A)X,$$

with $u$ the $N_c \times 1$ columnwise (temporal) mean vector of $A$ and $cov(A)$ is the covariance matrix of $A$. $b$ is an arbitrary scalar and can be set to $b = mean(I_{SoS})$ so that after optimization the mean temporal image is conveniently the same as for SOS. As a result, the optimization problem is a quadratic program under a linear constraint whose solution satisfies the Karush–Kuhn–Tucker (KKT) conditions [18]. The Lagrangian multiplier method yields for the Lagrangian

$$L(X, \lambda) = X'cov(A)X + \lambda(u'X - b),$$

with $\lambda$ the Lagrange multiplier. The solution can be found by setting the derivatives of the

Lagrangian with respect to $X$ and $\lambda$ to zero such that

$$\begin{cases} \dfrac{\partial L}{\partial x} = 2\Psi_t X + \lambda u = 0 \\[2mm] \dfrac{\partial L}{\partial \lambda} = uX - b = 0 \end{cases}$$

The solution then is $X = -\frac{\lambda cov(A)^{-1}u}{2} = -\frac{\lambda \Psi_t^{-1}u}{2}$. $\lambda$ is equal to $\frac{-2b}{u'\Psi_t^{-1}u}$ to meet the constraint so that $X = \frac{b\Psi_t^{-1}u}{u'\Psi_t^{-1}u}$. Conveniently in this final expression b and the denominators are scalars and therefore can be potentially omitted with no impact on the tSNR. The STARC voxel intensity thus finally simplifies to $I_{STARC} = A\Psi_t^{-1}u$.

This expression has some similarities with the one of covSOS$_t$ in that both exploit the time-series covariance matrix, but they are not identical. This result can already provide hints on the behaviour of STARC: because it uses the inverse of the time-series covariance matrix $\Psi_t^{-1}$, STARC will penalize the channels with the highest variability, regardless of its origin (neural activations or noise). We will experimentally show that this type of combination indeed greatly improves tSNR. But it comes at the cost of weaker t-scores because activation spikes are not dissociated from the rest of signal variability.

Therefore, in order to ignore activation-related signal fluctuations during the STARC optimization, a first simple solution would be to calibrate the weighting vector $X$ based on a separated in vivo pre-scan acquisition with no stimuli and apply it to the fMRI acquisition, assuming reproducibility. We will denote this coil combination STARC$_{ps}$.

A second alternative consists of replacing the tSNR as cost function by the t-score for the mean described by Corbin et al. [11], yet still neglecting temporal correlations to preserve an analytical solution for simplicity. The t-score for the mean of a voxel is the ratio of the temporal mean of the signal by the standard deviation of its residual after the General Linear Model fit (GLM). For clarity, the key aspects are reminded here. The detection of BOLD signal in fMRI exploits the General Linear Model [19], which describes the voxel-wise fMRI signal $I$ with a design matrix $D$ containing the explanatory variables such that

$$I = D\beta + \varepsilon,$$

with $\beta$, the regression coefficients estimated via least squares method and $\varepsilon$ the residual of the estimation assumed to follow a centred Gaussian distribution. Their respective expressions are

$$\beta = (D'D)^{-1}D'I = D^+I, \;\; \varepsilon = I - D\beta = (Id - DD^+)AX = PAX,$$

where $Id$ is the identity matrix and $D^+$ the pseudo inverse of $D$. The matrix $P$ defines a projection because $PP = P$. The typical method to infer on neuronal activations uses t-score statistics to assess the significance of given explanatory variables on the fMRI signal. The t-score is calculated with

$$t = \frac{c'\beta}{\sigma_\varepsilon \sqrt{c'(D'D)^{-1}c}},$$

where $c$ is a vector selecting a specific contrast. $\sigma_\varepsilon^2$ is the variance of $\varepsilon$ computed as

$$Var(\varepsilon) = \sigma_\varepsilon^2 = \frac{1}{N}X'\big((PA)'PA - u_{PA}u_{PA}'\big)X = X'cov(PA)X = X'\Psi_{tsm}X,$$

with $u_{PA}$ the column-wise (temporal) mean of PA. By the same token, just like for the STARC

optimization, consequently optimization of the t-score is written as

$$X^* = \underset{X}{\operatorname{argmin}} \ Var(\varepsilon) \ s.t. \ c'\beta = c'D^+AX = b,$$

where $b$ is again an arbitrary scalar. The t-score for the mean metric makes use of the expression of $t$ where $c$ is made of zeros everywhere except one at the index corresponding to the column of the design matrix fitting the mean of the signal (hence the name "t-score for the mean"). The t-score for the mean as a result can be interpreted as a tSNR where the standard deviation of the signal is calculated in the vector space orthogonal to the subspace spanned by the columns of $D$ (i.e. its image). That way, noise evaluation elegantly disregards what is believed to be activations. The function to maximize here is the t-score for the mean, the solution is found to be $X = -\frac{\lambda cov(PA)^{-1}c'D^+A}{2} = -\frac{\lambda\Psi_{tsm}^{-1}u_{tsm}}{2}$ through the same Lagrangian multiplier method as for the STARC optimization. Finally, the expression for the coil combination optimizing for the t-score for the mean yields for image intensity

$I_{STARC_{tsm}} = A\Psi_{tsm}^{-1}u_{tsm}$ (omitting again scalar coefficients with no impact on the t-scores). The pseudo-code for the t-score of the mean optimization is provided in S1 File.

Because this strategy actively uses the GLM design matrix $D$ for denoising, the degree of double-dipping [20] to which STARC$_{tsm}$ is prone has to be determined. t-scores normally follow the Student's t-distribution, but in the case of a high number of degrees of freedom (>30), this distribution is in practice set to be equivalent to a standard normal distribution of mean zero and standard deviation of one when there is no activation (null hypothesis). Double-dipping bias will be confirmed if STARC$_{tsm}$ does not follow this distribution.

## Materials & methods

### Sequence, pulse design and fMRI paradigm

This study was approved by the local ethics committee and four healthy volunteers were scanned after providing written informed consent. In vivo experiments were performed on a Magnetom 7T (Siemens Healthineers, Erlangen, Germany) with software version VB17A step 2.3 and equipped with a parallel transmission (pTX) 8Tx/32Rx head coil (Nova Medical, Wilmington, MA, USA).

For each volunteer, anatomical scans were performed with a 0.8 mm isotropic resolution MPRAGE sequence and functional scans were acquired with 3D-EPI [21]. Imaging parameters for the MPRAGE were TR/TI/TE 2600/3.44/1100 ms, flip angle (FA) 4˚, 192 sagittal slices, FOV 256 mm, acceleration factor 2 with GRAPPA. Parameters for the 3D-EPI were chosen to be as close as possible to the ones from the 7T Human Connectome Project (HCP) resting-state fMRI protocol. Parameters were: TR/TE 55/22 ms (TRvol = 1.2 s), 88 sagittal slices, 1.6 mm isotropic resolution, FOV 208 mm, acceleration factor 2×4, partial-Fourier (PF) 7/8, CAIPI shift $\Delta k_z$ = 2, BW = 1832 Hz/Px, Posterior to Anterior phase encoding, total acquisition time 5:26 mins for 260 repetitions. A brief 3D-EPI scan with the same parameters was acquired with an inverted phase encoding (Posterior to Anterior) for distortion correction.

To improve the RF transmit field, universal pulses [8, 22] were specifically designed for the 3D-EPI sequence via an offline optimization algorithm, using a database of 20 $B_1$/$B_0$ field maps. Flip-angle normalized root mean square error was reduced from 21% in CP-mode to 12% with universal pulses. The optimized pulse was a three $k_T$-points RF pulse of flip-angle 15˚ and 3 ms total duration for water selection [8].

In order to assess the differences in BOLD sensitivity between the different RF coil combinations, a localizer fMRI paradigm was used [23]. It is a fast event-related fMRI paradigm

consisting of a succession of ten types of stimuli such as checkerboard, auditory/visual sentences, calculations and right/left clicks.

## Data analysis and comparison strategy

The 3D-EPI data were reconstructed with a custom GRAPPA reconstruction code and yielded uncombined channel images. The three dimensional time series obtained after each coil combination were normalized so that their mean temporal signal was set equal to the mean temporal of SoS and then saved into NifTI format. Image alignment, 2 mm$^3$ FWHM Gaussian spatial smoothing and brain normalization on 2mm$^3$ MNI were done with SPM12 [24]. A distortion correction was performed with topup [25] from the FSL library (FMRIB, Oxford UK) between image realignment and registration steps. The GLM design matrix included task onsets convolved with a canonical hrf with their first derivative, signal drift, mean and six motion regressors (three translation, three rotations) obtained from the motion compensation post-processing step (after coil combination).

For all volunteers, we implemented the covSoS, STARC, STARC$_{ps}$ and STARC$_{tsm}$ combinations from the same acquisitions to ensure a fair comparison. Only the coil combination differed. An additional acquisition with the same sequence and parameters of 100 repetitions with RF but without any stimuli was performed in order to obtain optimised weights; the weights were then applied on the uncombined fMRI data to get the STARC$_{ps}$ reconstruction. tSNR maps were computed by taking the ratio of the temporal mean to the standard deviation over the 260 volumes. For each coil combination, boxplots of the tSNR distributions pooled on the four volunteers were also computed with a brain mask to exclude non-brain voxels. By fitting the post-processed data with the GLM, SPM returned t-score maps as indication of activation strength. Apart from visual qualitative analysis to ensure no absurd activated clusters, quantitative analysis consisted in the evaluation of the number of activated voxels for different p-values and for each coil combination strategy. In order to analyze the link between tSNR and t-score, we generated a scatter plot of the tSNR ratio STARC/covSoS versus the t-score ratio STARC/covSoS. Likewise, to determine whether an increase in t-score for the mean is sufficient to get an increase in t-score, we also generated a similar plot of the t-score for the mean ratio STARC/covSoS versus the t-score ratio STARC/covSoS. Activated voxels from motor and auditory contrast activation maps (covSoS) were considered. A regression coefficient was computed for each plot in which all points were pooled together. Monte Carlo simulations were performed to evaluate the tendency of STARC$_{tsm}$ to double-dipping. 32 Gaussian random noise waveforms with 260 points each were generated. Their mean was uniformly and randomly distributed between 0 and 50 and the noise level was set to be 5% of the weakest signal mean. No correlations among the channels were imposed. The null hypothesis for t-score testing assumes no activation; therefore, no activations were added to the waveforms. These signals were combined with the covSoS method and STARC$_{tsm}$, as presented in the Theory section. The thermal noise covariance matrix input was the noise level squared times the identity matrix. Simulated covSoS and STARC$_{tsm}$ signals were then fitted with the GLM and a t-score was calculated for a single task contrast and saved. The design matrix used was the same as the one used for the in vivo experiment excluding motion regressors. This process was repeated 10$^5$ times to yield a smooth t-score probability density function for covSoS and STARC$_{tsm}$. In order to understand how STARC sets the weights for each channel, the mean, the standard deviation and the optimized weights for each channel were displayed by means of bar plots for two activated voxels. Scatter plots of the mean versus the standard deviation across receive channels were also displayed for each voxel, a linear fit including the ten strongest channels was plotted on top of them. Finally, the time series of two activated voxels

(according to auditory and visual covSoS contrast maps p<0.001) were plotted for covSoS and STARC coil combinations. All time series had their mean previously normalized to allow a fair comparison. The time-series of the stimuli onsets convolved with the canonical hrf was like-wise displayed to ease the distinction between noise and activations.

## Results

Before exploring the results, we briefly remind for clarity the key principles of each coil combination. STARC is a weighted sum of channels optimized to yield the best tSNR. The tSNR of a voxel is defined here by the ratio of its temporal mean and its standard deviation. Because neuronal activations have also an influence on the tSNR, two other modified STARC versions were investigated. $STARC_{ps}$ consists in optimizing the weights on a task-free scan and then applying them to the fMRI scans. Finally, $STARC_{tsm}$ is a weighted sum of channels optimized to yield the best t-score for the mean. The t-score for the mean is similar to tSNR but normalized by the standard deviation of the residuals of the GLM. Believed neuronal activations are thus not included as noise.

Fig 1 reports the tSNR returned from covSoS, STARC, $STARC_{ps}$ and $STARC_{tsm}$ coil combinations. STARC showed the best overall tSNR performance up to a twofold tSNR increase compared to the gold standard covSoS. $STARC_{tsm}$ also greatly improved tSNR but not as much as STARC, indicating the influence of the potential activations on the tSNR evaluation. On the other hand, $STARC_{ps}$ did not improve tSNR at all compared to covSoS which shows that the optimized weights can be scan dependent.

Fig 2 displays activation maps, from the same volunteer as in Fig 1. While STARC yielded the highest tSNR, it performed poorly compared to the covSoS in terms of BOLD detection. Its activation map has smaller clusters and weaker t-scores. $STARC_{tsm}$ (without correction of the statistical bias induced by double-dipping) apparently improved the detectability since potential activations were removed before optimization. Given the fact that $STARC_{tsm}$ yields a

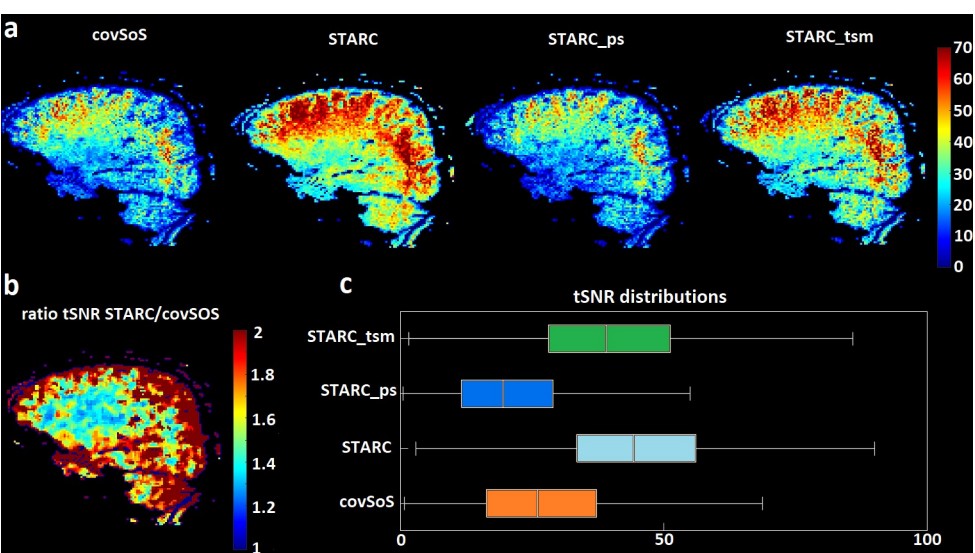

**Fig 1. tSNR results.** a–tSNR map from the same volunteer, scan and sagittal slice after covSoS, STARC, $STARC_{ps}$ and $STARC_{tsm}$ coil combinations. STARC yielded the highest tSNR map. b–Ratio of STARC and covSoS tSNR maps after median image filtering. STARC always improves tSNR, up to a factor 2. c–tSNR distributions pooled across all volunteers. Overall, STARC outperforms the other coil combinations in terms of tSNR. A brain mask was used to ignore non brain voxels.

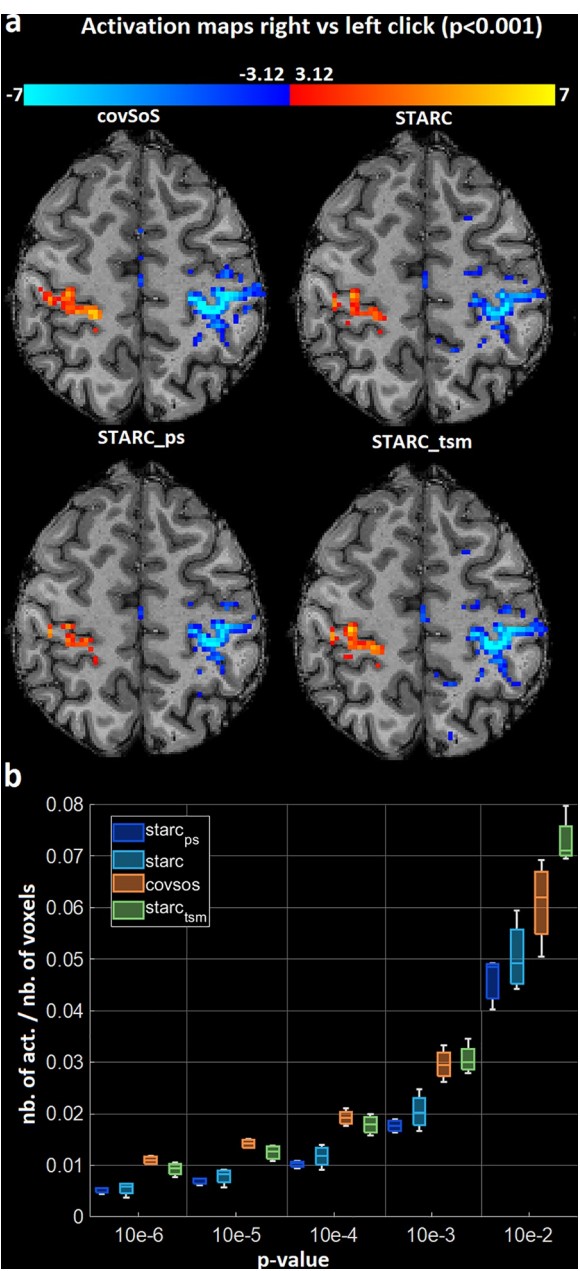

**Fig 2. Activation results.** a–Activation maps for the motor contrast at p<0.001 (no correction) for covSoS, STARC, STARC_ps and STARC_tsm. No double-dipping correction was applied to STARC_tsm. The shown maps are from the same volunteer as in Fig 1. b–Total number of activated voxels relative to the total number of voxels at different p-values and for different coil combinations, pooled over the volunteers. STARC_ps and STARC have the poorest performance in terms of BOLD detection. CovSoS has the highest number of activations for the lowest p-values but STARC_tsm (without double-dipping correction) reports more activations for the highest p-values.

slightly smaller tSNR, we see that part of the preserved variations is linked to neuronal activity. STARC_ps shows again the lowest performance confirming that calculated weights here are irrelevant if used on different scans. Moreover, since the method is a voxel-based optimization, inter-scan motion can also make the weights suboptimal.

Fig 3 shows for each activated voxel the effect of tSNR improvement on BOLD detection. All plotted voxels have an improved tSNR thanks to STARC coil combination but their t-score

is for most of them reduced. The regression coefficient of -0.41 suggests that tSNR not only is poorly correlated with t-scores but also indicates a negative trend when using such data-driven optimization approaches. An increase of tSNR with STARC can lead to an increase of t-score for the mean, though non-linear [11]. Yet again, Fig 3B shows in this case poor correlation between t-score for the mean and t-scores, when using the proposed strategy. The few voxels where the tSNR is reduced after STARC are likely due to the combination of no tSNR improvement and successive post-processing operations that slightly reduced the tSNR.

Fig 4 shows that the null hypothesis of STARC$_{tsm}$ seems to have a distorted distribution compared to covSoS with a higher variance. The distortion is significantly different to covSoS ($p < 0.05$). A simple solution consists in dividing the t-statistics of STARC$_{tsm}$ by the standard deviation characterized here under the null hypothesis, to enforce a standard normal distribution. This double-dipping correction reduced the number of activated voxels by almost a factor of 2.

Fig 5 shows by means of bar plots for two different voxels located in grey matter, how distributed the weights are across channels depending on the signal strength. Because there is a clear correlation between signal strength and signal variability, the STARC approach promotes channels with weak variance and thus of weak signal. Moreover, if the signals between two

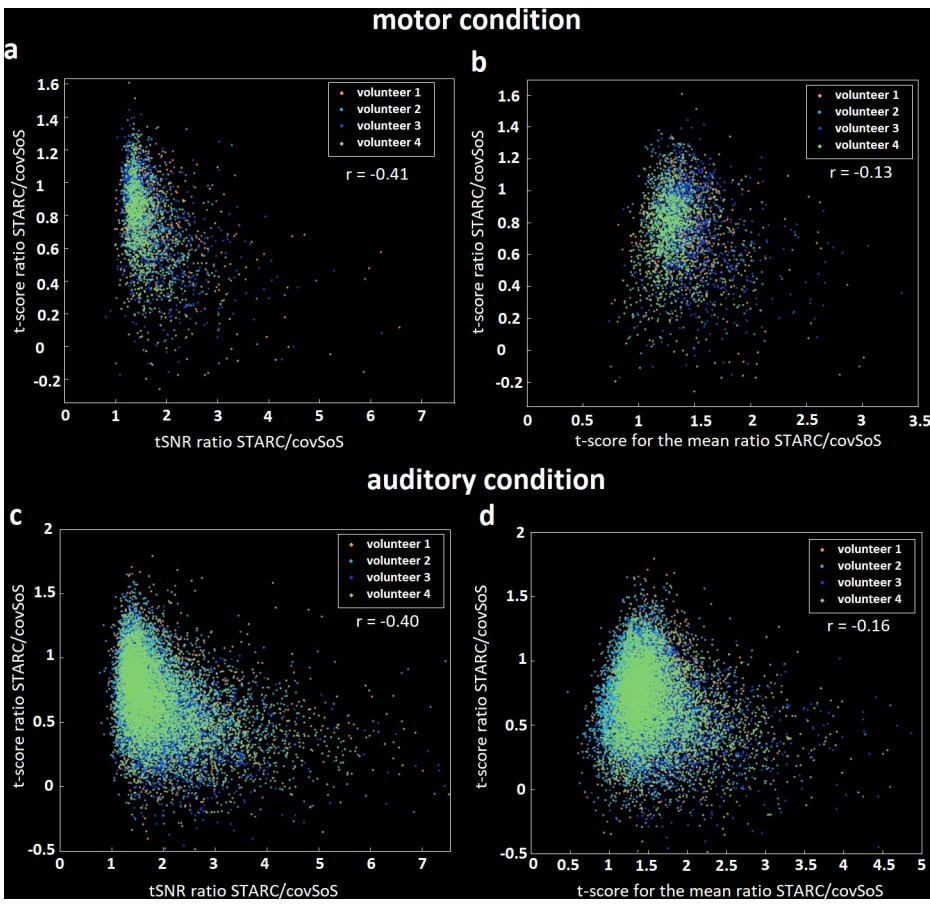

**Fig 3. t-score gains versus tSNR or t-score for the mean gains.** a—Scatter plot linking the activated voxels, the tSNR gain with STARC to its t-score gain for motor contrast. b–Scatter plot linking the t-score for the mean gain with STARC to its t-score gain for motor contrast. c–Same plot than a but with auditory contrast. d–Same plot than b but with auditory contrast. Each point corresponds to an activated voxel ($p<0.001$) according to the covSoS combination maps. In general, an increase of tSNR did not yield a better t-score.

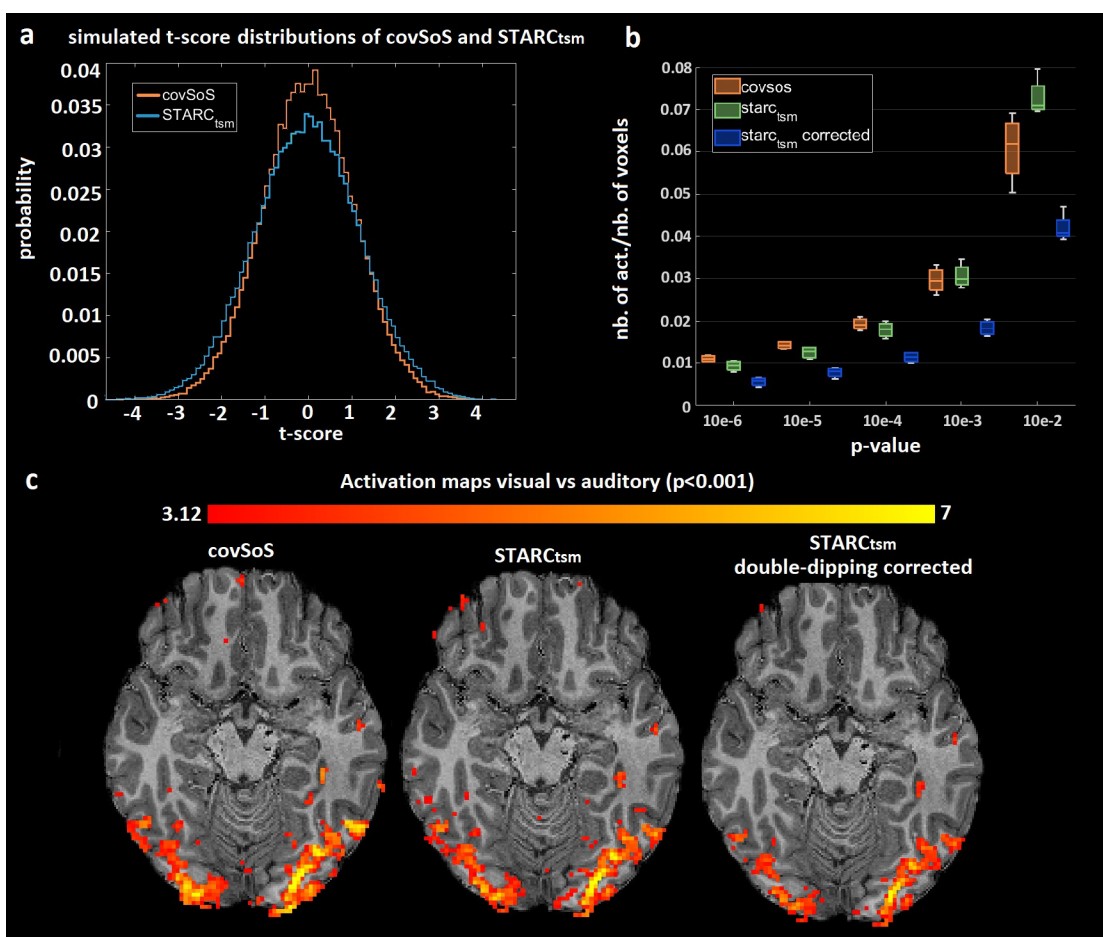

**Fig 4. Double-dipping results.** a–T-score distributions of covSoS and STARCtsm from noise signals and Monte Carlo simulations. The distribution of STARCtsm under the null hypothesis is distorted compared to covSoS because of double-dipping. b–Total number of activations relative to the total number of voxels at different p-values and for covSoS, STARCtsm and its double dipping corrected version for the in vivo scan. After correction, the number of activations from STARCtsm markedly drops. c–Activation maps on one volunteer for visual vs auditory contrast at p<0.001 (no correction). After double-dipping correction, activations are weaker and clusters smaller.

channels are highly correlated, e.g. when there are activations, then STARC can subtract one to another to remove them. The linearity between mean and standard deviation in the scatter plots favours the physiological noise regime hypothesis because thermal noise is not proportional to signal strength. Here, the slope of the fit corresponds to a rough estimation of 1/tSNR. However, we see that the weakest channels have their respective points not aligned with the others suggesting that the weaker the signal, the more the noise will be in the thermal regime.

Fig 6 plots the time series from covSoS and STARC combinations from the same activated voxels. covSoS time series yield the highest activation peaks but also has the highest variability. In the end, even if STARC has the highest tSNR (tSNR covSoS/STARC 35.4/67.6 and 67.2/85.0 for the first and second voxels respectively), the t-scores will be smaller than for covSoS (t-scores covSoS/STARC are 14.1/9.1 and 11.8/8.5 for the first and second voxel respectively). The chosen voxels here were strongly activated, which translates into very high activation peaks. These peaks representing the bulk of signal variability are thus reduced to increase tSNR with STARC.

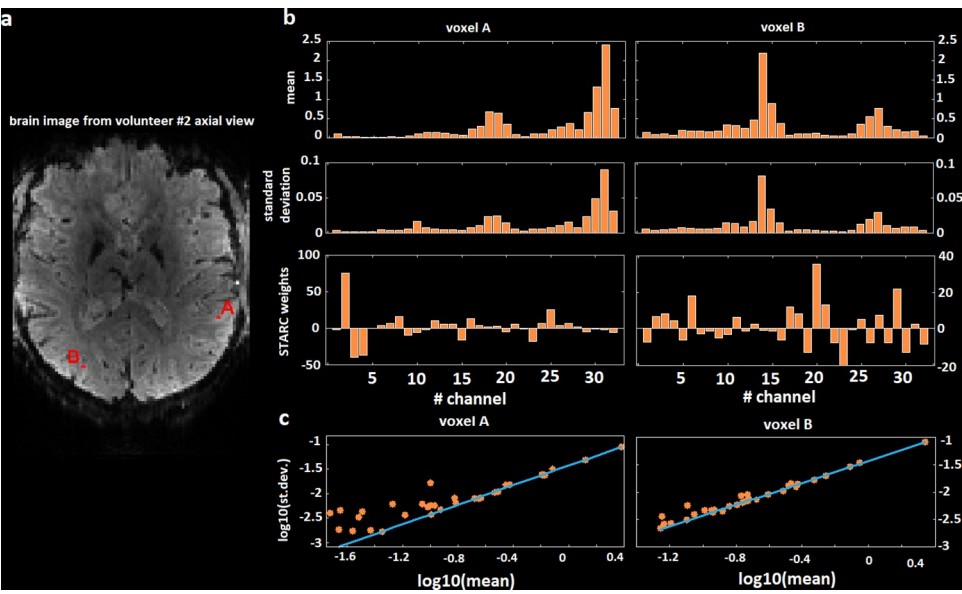

**Fig 5. Weights distributions from STARC optimization for two voxels.** a–Brain image from volunteer #2 axial view, the red points labelled A and B are the voxels whose weights are analysed. b–For both voxels A and B, the bar plots display the temporal mean, the temporal standard deviation and the optimized STARC weights. The channels receiving the strongest signal have the highest variability but will be given the lowest weight. c–Scatter plot of the mean versus the standard deviation for each channel, the slope of the linear fit of the ten strongest channels is also plotted. The signal variability is highly proportional to the signal strength.

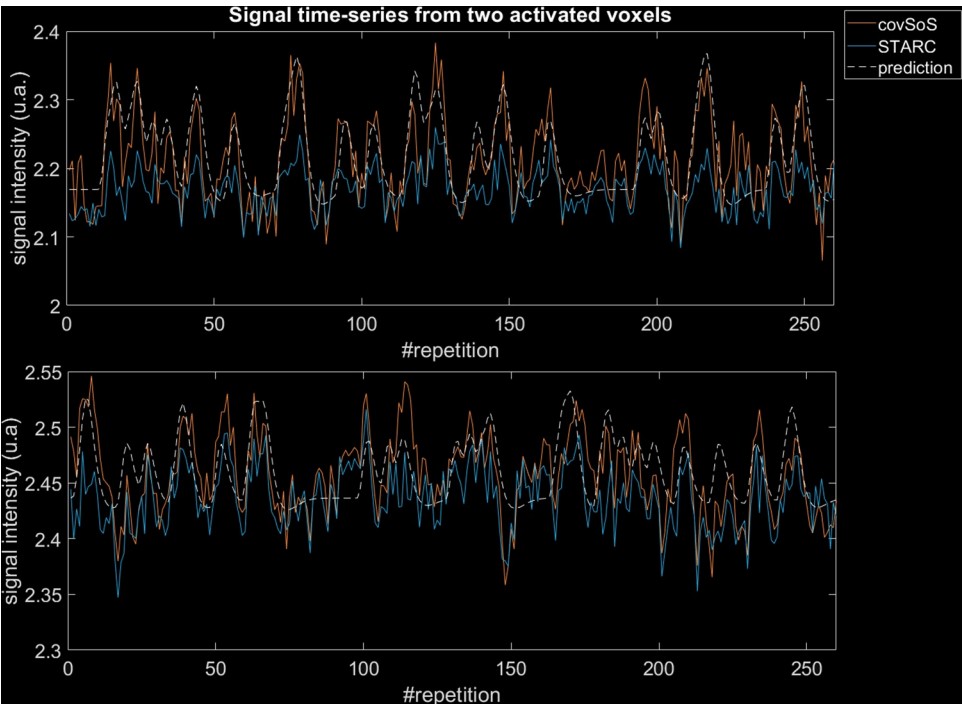

**Fig 6. Signal time series for two activated voxels for covSoS and STARC.** A time series of the stimuli onset convolved with the canonical hrf is also displayed. Each graph corresponds to a voxel. CovSoS has higher activation peaks than STARC but the latter has the highest tSNR.

## Discussion

Motivated by the stagnation of tSNR despite the increase in SNR in the presence of scanner instabilities and/or physiological noise, and by the preference of tSNR over SNR as quality metric in fMRI, we investigated in this work tSNR and t-score for the mean optimality through voxel-wise data-driven coil combinations.

We showed that optimality could be reached for each voxel in the image by providing the right formulation of the optimization problem. Minimizing the temporal variance of the resulting signal while having its mean equal to an arbitrary constant yielded an analytical solution that uses the coil-to-coil (total) noise covariance matrix. Despite a great improvement of tSNR, activation maps obtained after STARC coil combination were not as good as those from covSoS. Indeed, signal variation contains noise but also neuronal activations. By reducing the variations, noise was reduced but activation spikes were reduced too. By looking closer at the coil combination weights, STARC penalizes the coils with the highest signal and promotes the weakest ones because the physiological noise or activations are most of the time proportional to signal strength while thermal noise can be roughly constant across all coils. This means that the STARC signal for a given voxel location will be mostly made from the receive coils that are the farthest to that location. Moreover, the poor results in tSNR and t-score from the pre-scan strategy (STARC$_{ps}$) show that the optimization can be scan-dependent despite the use of the same sequence with same settings. The covariance matrix of the signals across channels ($\Psi_t$) indeed can be sensitive to the particular signal instance or random noise sample. Unless the time-series are particularly long, the covariance matrix calculated with modestly long time series will exhibit fluctuations. This gives to the STARC approach an "opportunistic" behaviour when it comes to reduce the signal variability. The most extreme scenario would be a time-series of length $N_c$, where STARC would return infinite tSNR (assuming linear independence of the $N_c$ signals), i.e. no signal variation. Applying the same optimized weights on another time-series would not return the same result because of a different noise sample.

The STARC method was unable to distinguish physiological noise from neuronal activity without any a priori while no assumptions about the nature of the noise or its statistics here were made. This motivated the change of strategy by optimizing the t-score for the mean instead of tSNR. Mathematically, it was shown to be a projection of the STARC problem onto a space orthogonal to the one spanned by the design matrix of the fMRI experiment. By doing so, potential activations were removed prior to variance minimization and an analytical solution again could be obtained. This strategy allowed having more activations than STARC but, as the Monte-Carlo simulations showed, this was an illusory consequence of double-dipping [20]. Once the null-hypothesis was characterized properly, the necessary corrections revealed no gain compared to covSOS. The optimization recipe provided in fact can be applied for any contrast optimization, but would lead again to double-dipping.

The t-score results of STARC do not follow those from the original abstract [16] but they confirm the results from a pilot experiment [17] where STARC returned also lower t-scores at higher resolution acquisitions, i.e. more in the thermal noise regime. The discrepancy could be temptingly explained by the fact that the original STARC algorithm consisted in a gradient descent with maximum and minimum constraints on the weights. These constraints could have possibly alleviated the destructive effects of the approach on the activations. Since an analytical solution was obtained, we wanted in this study to not include such constrains but rather emphasize the risks of searching for the best tSNR possible in data-driven optimization approaches. Despite the agreement between our observations and the results described in [17], we do not rule out the possibility of obtaining more activations in some other fMRI protocols based on such data-driven optimization principles.

In their paper [15], Triantafyllou et al. extends the KG-model by adding a new term to its expression. They define it theoretically as not scaling with signal strength. They confirm experimentally its physiological noise origin by showing that the value of this term was higher on in vivo than on phantom acquisitions. They propose examples of noise that do not scale with signal strength. These being seen simultaneously across channels, it can also presumably be reduced with linear combinations as in STARC. In our case, the strong proportionality between signal strength and signal variability shown in Fig 5 suggests that this type of noise here did not have an important influence on the STARC weights.

tSNR can increase if the signal mean increases (with constant or slower increase of variance) or if the variance decreases (with constant or slower decrease of mean). If the tSNR is computed on acquired fMRI scans, both cases do not necessarily represent a favourable situation to see more activations, because in the first case, the physiological noise is most of the time proportional to signal temporal mean [7, 15] and in the second case minimizing all signal variations can erode activation spikes. Although tSNR still remains an interesting metric to benchmark sequence settings, these results shed light on its limits. The results of this work suggest that tSNR should not be considered alone when evaluating or comparing acquired in vivo data because it contains neuronal fluctuations. The t-score for the mean is an interesting alternative to tSNR that does not take into account neural activity. However, we showed that an increase in the t-score for the mean is neither a necessary nor a sufficient condition to get an increase in t-score, but it depends on the strategy with which the t-score for the mean increase is obtained [11].

In conclusion, we presented tSNR and t-score for the mean optimal coil combination methods. Despite the proven optimality of these coil combinations for these data quality measures, activation maps did not improve compared to the gold standard covSoS coil combination, thereby indicating potential limits of tSNR and t-score for the mean metrics when assessing the quality of already acquired fMRI data. Although their approximate nature had already been reported in the literature, this work emphasizes the pitfalls associated with data-driven optimization approaches. Moreover, introducing prior information about the activations through the use of the GLM design matrix failed to outperform the covSoS coil combination method, once the statistical bias was taken into account properly. Although the theory leading to covSoS does not incorporate non Gaussian temporal fluctuations related to scanner instabilities or physiological noise, it appears that it remains so far more reliable for fMRI than data-driven coil combination methods being able to boost tSNR.

## Supporting information

**S1 File. STARC and STARCtsm pseudo code algorithms.**
(PDF)

## Acknowledgments

We thank Laurentius Huber for valuable discussions. We also thank the reviewers for their valuable comments.

## Author Contributions

**Conceptualization:** Redouane Jamil, Alexandre Vignaud, Nicolas Boulant.

**Data curation:** Redouane Jamil.

**Formal analysis:** Redouane Jamil, Caroline Le Ster, Nicolas Boulant.

**Funding acquisition:** Nicolas Boulant.

**Investigation:** Redouane Jamil, Nicolas Boulant.

**Methodology:** Redouane Jamil, Franck Mauconduit, Nicolas Boulant.

**Project administration:** Redouane Jamil, Nicolas Boulant.

**Resources:** Philipp Ehses, Benedikt A. Poser, Nicolas Boulant.

**Software:** Redouane Jamil, Franck Mauconduit, Caroline Le Ster, Philipp Ehses, Benedikt A. Poser, Nicolas Boulant.

**Supervision:** Nicolas Boulant.

**Validation:** Redouane Jamil, Nicolas Boulant.

**Visualization:** Redouane Jamil.

**Writing – original draft:** Redouane Jamil, Franck Mauconduit, Caroline Le Ster, Philipp Ehses, Benedikt A. Poser, Alexandre Vignaud, Nicolas Boulant.

**Writing – review & editing:** Redouane Jamil, Franck Mauconduit, Caroline Le Ster, Philipp Ehses, Benedikt A. Poser, Alexandre Vignaud, Nicolas Boulant.

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
