## [Decision Letter · Decision Letter 0]

6 Aug 2021

PONE-D-21-17255

Temporal SNR optimization through RF coil combination in fMRI: How high can we go?

PLOS ONE

Dear Dr. Boulant,

Thank you for submitting your manuscript to PLOS ONE. After careful consideration, we feel that it has merit but does not fully meet PLOS ONE’s publication criteria as it currently stands. Therefore, we invite you to submit a revised version of the manuscript that addresses the points raised during the review process.

We look forward to receiving your revised manuscript.

Kind regards,

Xi Chen

Academic Editor

PLOS ONE

Journal Requirements:

The research leading to these results has received funding from the ERPT equipment program of the Leducq Foundation. This project also has received funding from the European Union’s Horizon 2020 research and innovation program under grant agreement No 885876 (AROMA project). We thank Samy Strola of Absiskey for support in project management.

Leducq foundation

large equipment ERPT program,NEUROVASC7T project

https://www.fondationleducq.org/

European Union’s Horizon 2020 research and innovation program (AROMA project)

Grant no 885876

https://aroma-h2020.com/

NO - The funders had no role in study design, data collection and analysis, decision to publish, or preparation of the manuscript.

Reviewers' comments:

Reviewer's Responses to Questions

**Comments to the Author**

1. Is the manuscript technically sound, and do the data support the conclusions?

Reviewer #1: Partly

Reviewer #2: Partly

2. Has the statistical analysis been performed appropriately and rigorously? 

Reviewer #1: Yes

Reviewer #2: Yes

3. Have the authors made all data underlying the findings in their manuscript fully available?

Reviewer #1: No

Reviewer #2: No

4. Is the manuscript presented in an intelligible fashion and written in standard English?

Reviewer #1: Yes

Reviewer #2: Yes

5. Review Comments to the Author

Reviewer #1: The authors have presented an investigation into the impact of several RF receive array coil combination strategies on temporal signal-to-noise ratio (tSNR) and functional MRI detection sensitivity. This included an analysis of a previously proposed voxel-wise, data-driven tSNR-weighted combination, termed STARC, and variants of this technique that the authors proposed based on a pre-scan estimation of the STARC coil combination weightings and a related method based on the t-score of the mean. The authors found that, despite large improvements in tSNR, most of the STARC techniques had no benefit or and even reduced the detection sensitivity to their stimulus.

Very often, tSNR is the primary metric for assessing new methods in fMRI, but the authors argue here that optimizing tSNR is not necessarily entirely reflective of fMRI detection sensitivity. I think this is a valid point, although I am not convinced based on the experiments and analyses performed here that the authors’ conclusions are entirely accurate. I think that with additional experimental data, these conclusions could be much more well-validated and that with alternative, more common, approaches to analyzing the same data, the results may change significantly. I don’t think acquiring new data is necessary for the manuscript, but it could significantly increase its impact. Without new data, the conclusions will need to be significantly tempered down in terms of scope of applicability. For these reasons, I suggest a major revision for this manuscript.

Major comments:

1. The authors have not characterized the noise regime, physiological or thermal noise dominated, that their protocol is operating in. Given that the focus of the paper is testing alternative coil combination strategies that can potentially reduce physiological noise and temporal instability, this is a significant omission. In Fig. 1, the covSoS tSNR values are in the range of ~0–70, this would appear to be thermal noise dominated. Many of the conclusions in the manuscript are quite strong although it seems like the conclusions could differ if the data were in a different noise regime and not just acquired under the single protocol used in this study. Ideally, one would like to see the impact of the different coil combinations under a range of SNR conditions (e.g., voxel size, flip angle, echo-time variation) as is commonly done in similar papers, e.g., refs 6, 7, 9, 15. In lieu of acquiring new data under varying protocols, can the authors elaborate on how the noise regime might impact the different STARC coil combinations compared to covSoS and reframe their conclusions in terms of their particular acquisition?

2. To calculate tSNR for the coil combination and for the analysis of the coil-combined data, the authors have taken the straightforward ratio of the signal mean over the standard deviation in their task fMRI data. A more appropriate and common calculation of tSNR in task data is based on normalizing by the standard deviation of the residuals of the GLM, as is described, for example, in Murphy et al. (ref. 14) and Corbin et al. (ref. 11). When the desired signal fluctuations are properly modelled in the GLM, this should ensure that they are not included as noise. Ideally, this may then provide the same coil weightings as the STARC_ps method. The authors should consider combining their data using this more common tSNR calculation. Also, since the GLM would have to be run twice, it would make sense to consider possibly inflated statistics, like what was performed for STARC_tsm.

3. The authors claim in the Discussion that the STARC method penalizes high-intensity voxels and promotes weak-intensity voxels (pg. 12, lines 324–329). This seems to be the meat of the problem (and likely related to my comment above) and it is surprising that this is only mentioned in the Discussion. While this may be intuitive for the authors, it would be great to see this explored more quantitatively in the Results with some concrete examples. Is this apparent in the resulting time series? This is in…stark contrast…with the originally presented method from Huber et al., where tSNR and z-statistics both increased significantly. The authors should discuss why the discrepancy may exist.

4. pg. 6, line 132: It’s not entirely clear to me how the authors arrived at the solution for the coil combination weighting matrix, X = –1/2 cov(A)^-1 u. This appears as if the authors arbitrarily set the Lagrange multiplier lambda to 1. I would expect the solution to have the parameter b in it. Considering that b is also somewhat of an arbitrary parameter, maybe this explains the solution, however, it was previously stated that b was constrained to be equal to the sum-of-squares mean value.

Minor comments

5. In STARC_tsm, my understanding is the GLM is run voxel-wise and per receive channel to estimate the t-score on the mean-based weights. Can the authors explain if they applied typical preprocessing steps (e.g., motion correction, detrending, smoothing, etc.) to every channel’s image and repetition prior to coil combination and why or why not?

6. Can the authors please provide some more details on the pre-scan acquisition for the STARC_ps method? How many repetitions were acquired? Presumably, unlike the calculation of the thermal noise covariance matrix, RF excitation was enabled for this scan?

7. I think the subtitle of the manuscript, “How high can we go?” is not totally appropriate for this study since I don’t think the question was really answered. I would reconsider the subtitle for something that better reflects the experiments performed and the conclusions drawn.

8. pg. 4, line 79: The reference number for Triantafyllou et al. should be [15]

9. pg. 5, line 128-129: Please consider rephrasing, “The Lagrangian multiplier method for the yields for the Lagrangian”

10. pg. 6, line 154: I believe “projector” should be “projection”?

11. pg. 6, line 157: The final ‘c’ in the denominator should not be transposed.

12. pg. 11, line 295: Should be figure caption ‘b’, not ‘c’

13. pg. 12, line 324: seems like a word or two is missing from “noise is reduced but activation spikes too”

Reviewer #2: Summary

Jamil et al. investigate the effect of different data-driven coil combinations methods that either maximise tSNR or the t-score of the mean on the t-score of the activation and the activation cluster size. They show that contrary to when the idea was first proposed (Huber et al., 2017), STARC or any of its variants do not increase t-scores for activation, which is in line with previous, initial pilot experiments (Kashyap et al., 2018).

This study addresses an interesting topic, which has been discussed in the community, but not thoroughly investigated so far. In addition, including the effect of double-dipping here provides a valuable reference given the current popularity of data-driven denoising strategies, and I recommend the authors to at least make this part of their code available. I have a few questions regarding the interpretation of the results and the effect of physiological noise (below). Overall, this study provides a valuable addition to the literature.

Major points

Line 254ff and 329ff: You mention in the results and the discussion that the optimization strategy is highly scan-dependent. This is quite a surprising result. Do you have any more insights into why this is the case? Given the relatively smooth coil sensitivities, and if the location of physiological noise sources does not change drastically from one scan to the next, it seems that the driving factor behind this is still unclear. Maybe comparing the coil combination weights for these two scenarios would provide some hints?

Fig 3: Given that the result cannot be fully explained and the effect seem to be somewhat elusive, could you provide more data, for example in the form of scatter plots for the other participants and task conditions in the supplementary?

Line 272ff: Similar to my questions before, can you provide examples that quantify the interscan motion and its effect on the coil combination weights?

Line 325ff: Can you provide examples of coil combination weights to show how strongly STARC penalizes coils with highest signal?

Line 345ff: Triantafyllou et al. (2016) actually discuss the presence of physiological noise which does not scale with signal strength. Could you comment on why this may or may not play a role for your study.

In general, can you provide more details on what you include in your physiological noise definition, and its statistical properties? Given that signals from breathing or cardiac activity are noise by definition only (i.e. they cause ‘true’ changes for example in T2*), the rationale why different channels should pick up these signal differently, and in such a way that they can be avoided during coil combination while the activation signal remains untouched, isn’t fully clear.

Minor Points

Line 45ff: Please add relevant references (see also previous point).

Line 54: yet -> step?

Line 74: time-correlations -> maybe temporal correlations?

Line 79: [14] -> [15]

Line 98: For clarity, could you define ‘H’ as well?

Line 107ff: ‘by substituting to the scalar coefficient expressing the effective strength of the physiological noise a physiological noise covariance matrix’ -> ‘by substituting the scalar coefficient expressing the effective strength of the physiological noise with a physiological noise covariance matrix’

Line 237: Why for only one task?

Line 252: For readability, you could briefly summarize the ideas behind STARC, STARCps, and STARCtsm again, as not everyone might read (or remember) the theory section in detail.

Line 265: figure 1 -> Figure 1

Line 288: fig.3.b. -> Fig. 3b

Line 295: …gain. c – Scatter … -> … gain. b – Scatter …

I would also recommend to include the ISMRM abstract by Kashyap et al. into the discussion (2018), as this corroborates your findings.

References

Huber, L., Jangraw, D.C., Marrett, S., Bandettini, P.A., 2017. Simple approach to improve time series fMRI stability: STAbility-weighted Rf-coil Combination (STARC), in: Proc. Intl. Soc. Mag. Reson. Med. 25.

Kashyap, S., Fritz, F.J., Harms, R.L., Huber, L., Ivanov, D., Roebroeck, A., Poser, B.A., Uludağ, K., 2018. Effect of optimised coil-combinations on high-resolution laminar fMRI at 9.4T. Presented at the Proc. Intl. Soc. Mag. Reson. Med. 26.

Triantafyllou, C., Polimeni, J.R., Keil, B., Wald, L.L., 2016. Coil-to-coil physiological noise correlations and their impact on functional MRI time-series signal-to-noise ratio. Magnetic Resonance in Medicine 76, 1708–1719. https://doi.org/10.1002/mrm.26041

6. PLOS authors have the option to publish the peer review history of their article (what does this mean?). If published, this will include your full peer review and any attached files.

Reviewer #1: **Yes: **Avery J.L. Berman

Reviewer #2: No

---

## [Decision Letter · Decision Letter 1]

22 Oct 2021

Temporal SNR optimization through RF coil combination in fMRI: The more, the better?

PONE-D-21-17255R1

Dear Dr. Boulant,

We’re pleased to inform you that your manuscript has been judged scientifically suitable for publication and will be formally accepted for publication once it meets all outstanding technical requirements.

Kind regards,

Xi Chen

Academic Editor

PLOS ONE

Additional Editor Comments (optional):

Reviewers' comments:

Reviewer's Responses to Questions

**Comments to the Author**

1. If the authors have adequately addressed your comments raised in a previous round of review and you feel that this manuscript is now acceptable for publication, you may indicate that here to bypass the “Comments to the Author” section, enter your conflict of interest statement in the “Confidential to Editor” section, and submit your "Accept" recommendation.

Reviewer #1: All comments have been addressed

Reviewer #2: All comments have been addressed

2. Is the manuscript technically sound, and do the data support the conclusions?

Reviewer #1: Yes

Reviewer #2: Yes

3. Has the statistical analysis been performed appropriately and rigorously? 

Reviewer #1: Yes

Reviewer #2: Yes

4. Have the authors made all data underlying the findings in their manuscript fully available?

Reviewer #1: No

Reviewer #2: No

5. Is the manuscript presented in an intelligible fashion and written in standard English?

Reviewer #1: Yes

Reviewer #2: Yes

6. Review Comments to the Author

Reviewer #1: (No Response)

Reviewer #2: All my comments have been addressed. Three typos I noticed:

Line 54: I meant to replace yet with step (or remove the article before pre-whitening).

Line 94: An example? Or Examples?

Line 407: ... without any a priori (what?) while no assumptions ...

7. PLOS authors have the option to publish the peer review history of their article (what does this mean?). If published, this will include your full peer review and any attached files.

Reviewer #1: **Yes: **Avery J.L. Berman

Reviewer #2: No

---

## [Editor Report · Acceptance letter]

27 Oct 2021

PONE-D-21-17255R1 

Temporal SNR optimization through RF coil combination in fMRI: The more, the better? 

Dear Dr. Boulant:

I'm pleased to inform you that your manuscript has been deemed suitable for publication in PLOS ONE. Congratulations! Your manuscript is now with our production department. 

Kind regards, 

on behalf of

Dr. Xi Chen 

Academic Editor

PLOS ONE